# Liver X Receptor Inverse Agonist GAC0001E5 Impedes Glutaminolysis and Disrupts Redox Homeostasis in Breast Cancer Cells

**DOI:** 10.3390/biom13020345

**Published:** 2023-02-10

**Authors:** Asitha Premaratne, Charles Ho, Shinjini Basu, Ashfia Fatima Khan, Tasneem Bawa-Khalfe, Chin-Yo Lin

**Affiliations:** Center for Nuclear Receptors and Cell Signaling, Department of Biology and Biochemistry, University of Houston, Houston, TX 77004, USA

**Keywords:** liver X receptor, ligands, breast cancer, metabolism, glutaminolysis

## Abstract

Liver X receptors (LXRs) are members of the nuclear receptor family of ligand-dependent transcription factors which regulate the expression of lipid and cholesterol metabolism genes. Moreover, LXRs and their ligands have been shown to inhibit tumor growth in a variety of cancers. We have previously identified the small molecule compound GAC0001E5 (1E5) as an LXR inverse agonist and a potent inhibitor of pancreatic cancer cells. Transcriptomic and metabolomic studies showed that 1E5 disrupts glutamine metabolism, an essential metabolic pathway commonly reprogrammed during malignant transformation, including in breast cancers. To determine the role of LXRs and potential application of 1E5 in breast cancer, we examined LXR expression in publicly available clinical samples, and found that LXR expression is elevated in breast tumors as compared to normal tissues. In luminal A, endocrine therapy-resistant, and triple-negative breast cancer cells, 1E5 exhibited LXR inverse agonist and “degrader” activity and strongly inhibited cell proliferation and colony formation. Treatments with 1E5 downregulated the transcription of key glutaminolysis genes, and, correspondingly, biochemical assays indicated that 1E5 lowered intracellular glutamate and glutathione levels and increased reactive oxygen species. These results indicate that novel LXR ligand 1E5 is an inhibitor of glutamine metabolism and redox homeostasis in breast cancers and suggest that modulating LXR activity and expression in tumor cells is a promising strategy for targeting metabolic reprogramming in breast cancer therapeutics.

## 1. Introduction

Liver X receptors (LXRs) are members of the nuclear receptor superfamily of ligand-dependent transcription factors. The two LXR subtypes (LXRα and LXRβ) regulate the expression of genes involved in cholesterol, lipid, and glucose metabolism and inflammatory responses [1,2]. LXR activity can be modulated by lipophilic endogenous and synthetic ligands which function as agonists or inverse agonists [3,4]. In addition to their metabolic functions, LXRs have emerged as potential targets in cancer [5,6]. Studies have shown the anti-proliferative effects of LXR ligands on tumor cells in different types of cancer models [7,8]. To identify novel LXR ligands with inhibitory activity in cancer, we conducted a screen of a focus small molecule library of predicted LXR ligands in pancreatic cancer cells, and identified compound GAC0001E5 (1E5), which showed strong anti-tumor activity [9]. Characterization of 1E5’s mechanisms of action indicated that it functions as an LXR inverse agonist which also drastically reduced LXR protein levels following treatment. A well-known LXR inverse agonist SR9243 has been shown to inhibit the Warburg effect in cancer cells and induce apoptosis [10]. Follow-up transcriptomic and metabolomic studies revealed that 1E5 disrupts glutaminolysis and increases intracellular oxidative stress in pancreatic cancer cells [11].

Metabolic reprogramming is a hallmark of cancer [12]. Glutamine dependency and increased glutaminolysis are well established features of a number of malignancies, including breast cancers [13,14]. In breast cancers, hormone receptor-positive luminal breast cancers exhibit variable glutamine dependency, whereas HER2-positive and triple-negative breast cancers appear to be highly dependent on exogenous glutamine and increased glutamine metabolism for survival [15,16,17]. Based on our findings in pancreatic cancer cells, we posit that targeting LXRs with the novel ligand 1E5 will similarly disrupt glutamine metabolism and tumor cell proliferation and survival in breast cancers [9,11]. To test this hypothesis, we examined LXR expression in different breast cancer subtypes and determined the metabolic and anti-tumor effects of 1E5 in cellular models of hormone receptor-positive luminal A, endocrine-resistant, and triple-negative breast cancers.

## 2. Materials and Methods

### 2.1. TCGA Gene Expression Data

Raw counts of RNA sequencing data from tumors of The Cancer Genome Atlas (TCGA) Breast Invasive Carcinoma (BRCA) project and GTEx normal breast tissue were downloaded using TCGAbiolinks (2.12.6) in R (3.6.1). Log normalized TPM values were calculated using gene lengths obtained from human genome 38 (hg38) from Rsubread (1.34.7). Welch’s two-sample t-test from the ggpubr (0.2.5) package was used to compare the mean values of each group. Survival analyses between patients with high and low gene expression were conducted using the survival (3.1–12) and survminer (0.4.6) packages. The maxstat function from survminer was used to find the relevant biological cutoff between patients with high and low gene expression during survival analysis. TCGAbiolinks and Rsubread packages were obtained from https://bioconductor.org accessed on 21 November 2022. The ggpubr, survival, and survminer packages were obtained from https://cran.r-project.org/ accessed on 21 November 2022.

### 2.2. Treatments, Cell Lines, and Cell Culture

Novel LXR ligand GAC0001E5 was synthesized by Otavachemicals, Concord, ON, Canada. *Synthetic LXR agonist* GW3965 and glutaminase inhibitor BPTES were obtained from Cayman Chemical, Ann Arbor, MI, USA. All cell culture media and FBS were purchased from Thermo Fisher Scientific Inc., Waltham, MA, USA. MCF-7 and MDA-MB-231 cell lines were purchased from the American Type Culture Collection (ATCC), Manassas, VA, USA. A Tamoxifen-resistant MCF-7-TamR cell line was obtained from the Bawa-Khalfe Lab. MCF-7 cells were treated with 1 μM of 4-Hydroxytamoxifen (Selleckchem, Houston, TX, USA) for six months for acquisition of resistance to produce MCF-7-TamR. Furthermore, they were supplemented with 1 μM of 4-Hydroxytamoxifen to maintain resistance [18]. MCF-7 and MCF-7-TamR cell lines were cultured in DMEM media (Gibco #12430054). MDA-MB-231 cells were cultured in DMEM/F12 (Ham) (Gibco #11330032). Each media type was supplemented with 10% FBS (Gibco #26140079). Cell cultures were maintained in a humidified atmosphere of 5% CO_2_ at 37 °C.

### 2.3. Cell Proliferation Assays

Tetrazolium salt reduction (MTT), trypan blue exclusion and colony formation assays were used to measure cell proliferation and survival. For MTT assays, MTS reagent [(3-(4,5-dimethylthiazol-2-yl)-5-(3-carboxymethoxyphenyl)-2-(4-sulfophenyl)-2H-tetrazolium) (Promega, Madison, WI, USA #G3582)] was used to determine cumulative cellular metabolic activity, which reflects the total number of live cells in culture. At hour 0, 1 × 10^4^ cells/well of each cell line were seeded into separate 96-well plates (100 μL/well) and allowed 24 h to attach. The cells were treated with different concentrations of ligands at desired incubation times. Subsequently, cells were washed with PBS and 100 μL of media and 10 μL of MTS were added to each well. Incubation periods with MTS varied from 45 to 120 min [MCF-7 (60 min), MCF7-TamR (45 min), and MDA-MB-231 (120 min)]. Upon incubation, the absorbance was measured at 490 nm using a microplate reader (SpectraMAX M5, Molecular Devices, San Jose, CA, USA). These assays were performed in biological triplicates with technical quadruplicates. To further quantify the efficacy of novel LXR ligands in breast cancer cells, IC50 determinations were performed using MTT assays. At 0 h, cells were seeded at 1 × 10^4^ cells/well density and allowed to attach for 24 h. Media was replenished with an array of 0.01 to 100 mM ligand concentrations for each ligand (1E5 and 3A4) and incubated for another 72 h. After the incubation period, MTT assays were performed as described. Data were normalized to DMSO (control) OD value and the percentage viability was calculated for each concentration. These data were used to calculate IC50 values.

A trypan blue exclusion assay was used to determine the number of living cells following treatments. Cells were seeded in 6-well plates at 2 × 10^5^ cells/ well density. After allowing 24 h for the cells to attach, the media were replenished with the treatments and incubated for another 48 h. Cells were collected by trypsinization, resuspended with media, and stained using trypan blue (Gibco, Thermo Fisher Scientific Inc., Waltham, MA, USA, #15250-061) to count live cells.

For the colony formation assays, cells were seeded in 6-well plates at low densities of 5 × 10^2^ cells/well for MCF-7, MDA-MB-231, and 1 × 10^3^ cells/well for MCF7-TamR. After 72 h from seeding, the treatments were introduced (DMSO, GW (10 μM), and 1E5 (10 μM)) without changing media, and the plates were incubated for another 48 h. Media and treatments were replenished after 5 days from seeding and every 5 days until the 15th day. Thereafter, colonies were washed twice with PBS (VWR, Radnor, PA, USA, #K813) and fixed with 4% PFA (paraformaldehyde) (Sigma-Aldrich, St. Louis, MO, USA, #158127) solution for 10 min. Fixed colonies were then stained with 500 μL of 0.5% (*v*/*v*) crystal violet (Sigma-Aldrich, St. Louis, MO, USA, #CO775) for 10 min. After dyeing, the excess crystal violet was removed and washed twice with PBS. Plates were then allowed to dry overnight before counting.

### 2.4. Real-Time Quantitative PCR

Cells were seeded at 2 × 10^5^ cells/well density in 6-well plates. Treatments were introduced with fresh media at 24 h and incubated for another 48 h: DMSO (vehicle), GW, 1E5, and 3A4 in 10 mM concentrations. Total RNA was extracted using the RNeasy Mini Kit (Qiagen, Germantown, MD, USA, #74106). RNA was quantified and 1000 ng was used for cDNA synthesis (iScript cDNA synthesis kit, Bio-Rad, Hercules, CA, USA, #1725035). Diluted cDNA (1:10) was utilized for the setup. qPCR reactions were set up to be 10 μL in total volume (0.5 μL forward primer, 0.5 μL reverse primer, 1 μL nuclease-free water (Invitrogen, Thermo Fisher Scientific Inc., Waltham, MA, USA, #AM9906), 5 μL of SYBR Green (Applied Biosystems, Thermo Fisher Scientific Inc., Waltham, MA, USA, #A25742), and 3 μL of diluted cDNA). Reactions were performed in a Biosystems 7500 Fast Real-Time PCR system (Applied Biosystems, Thermo Fisher Scientific Inc., Waltham, MA, USA). Primer sequences of the genes used are listed here.

36B4-F-GTGTTCGACAATGGCAGCAT

36B4-R-GACACCCTCCAGGAAGCGA

SREBP1c-F-GGAGGGGTAGGGCCAACGGCCT

SREBP1c-R-CATGTCTTCGAAAGTGCAATCC

ABCA1-F-TGTGAGGCGGGAAAGACAGAG

ABCA1-R-AGCCCAAAGCACTCACCAGGA

ABCG1-F-CGATGAGCCCACCAGCGGC

ABCG1-R-ACCCCCTTGAGCGAGCCCTT

ACC-F-GCAGGTCACACGTCTCTTTAT

ACC-R-CCAGCCTGTCATCCTCAATATC

SCD-F-TTCAGAAACACATGCTGATCCTCATAA

SCD-R-ATTAAGCACCACAGCATATCGCAAGAA

FASN-F-ACAGGGACAACCTGGAGTTCT

FASN-R-CTGTGGTCCCACTTGATGAGT

GLS1-F-TTCCAGAAGGCACAGACATG

GLS1-R-GGCTCAGTACTCTTTCACCAG

GOT1-F-CAACTGGGATTGACCCAACT

GOT1-R-GGAACAGAAACCGGTGCTT

GOT2-F-GTTTGCCTCTGCCAATCATATG

GOT2-R-GAGGGTTGGAATACATGGGAC

GLUD1-F-AGGAATGACACCAGGGTTTG

GLUD1-R-TCAGACTCACCAACAGCAATAC

SLC7A11-F-TTTCTGCATCCACATTCCAA

SLC7A11-R-AACACCATCTGGCATTGTGA

### 2.5. Western Analysis of LXRβ Expression

Total protein extracts were used for Western blot analysis. Cells were seeded at 1 × 10^6^ cells/plate density in 10 cm plates. After 24 h, the media were replenished with treatments (DMSO (vehicle), GW, 1E5, and 3A4 in 10 μM concentrations). After 48 h of incubation, plates were washed twice with PBS, then 1 mL of ice-cold PBS was added and the cells were scraped. The collected suspension was centrifuged at 5000 rpm at 4 °C for 5 min. Pellets were incubated with 150 μL Lysis Buffer (RIPA supplemented with protease inhibitor [Roche Diagnostics, Indianapolis, IN, USA, #11836170001]) for 30 min at 4 °C. The cell lysates were centrifuged at 13,000 rpm at 4 °C for another 15 min. The proteins were collected and quantified using Bradford assay (VWR, Radnor, PA, USA, #E530). For 12% SDS-PAGE gel, 25 μg of total protein was added from each sample. The completed gel was transferred to nitrocellulose paper (Thermo Fisher Scientific Inc., Waltham, MA, USA, #88518) for Western blot analysis. Upon transfer, the blots were placed in a blocking buffer (5% milk in TBST) for 1 h. After blocking, the blots were incubated overnight with LXRβ (R&D Systems, Minneapolis, MN, USA #PP-K8917) and β-actin (Sigma-Aldrich, St. Louis, MO, USA, #A2228) monoclonal antibodies. The anti-mouse polyclonal antibody was used as a secondary antibody (GE Healthcare, Chicago, IL, USA #NXA931V). Protein signals were developed using Clarity^TM^ Western ECL (Bio-Rad, Hercules, CA, USA, #1705061). The chemiluminescence signals were recorded by LI-COR Odyssey Fc (LI-COR Biosciences, Lincoln, NE, USA, # OFC-0842).

### 2.6. Glutamine Dependency Assays

MTT assays were performed to determine glutamine dependency on the cell proliferation in each cell line. Cells were seeded in 96-well plates, as described in the MTT assay in Section 2.3. Cells were treated with DMSO and 1E5 under two different media conditions, one with glutamine-stripped media and the other containing the supplement. After 72 h of incubation, an MTT assay was performed.

### 2.7. Intracellular Glutamate Measurements

Intracellular glutamate levels were assayed using the Glutamate-Glo^TM^ Kit (Promega, Madison, WI, USA, #J7021). Two sets of cells were seeded for each cell line in two 96-well plates, one for the assay itself and another for MTT normalization. Both plates were seeded with 1 × 10^4^ cells/well density for each cell line, and the cells are allowed to attach for 24 h. In conclusion, both plates were treated with 6 different treatment sets [DMSO (vehicle), 1E5 (5 μM), 1E5 (10 μM), BPTES (5 μM), BPTES (10 μM) and 1E5 (5 μM) + BPTES (5 μM)]. Treatments were then incubated for 48 h and MTT assays were performed for one set of plates. The glutamate assay plate was washed with PBS twice, and 37.5 μL of inactivation solution (12.5 μL of 0.6 N HCl and 25 μL of PBS) was added and shaken for 5 min. The inactivation solution was neutralized using 12.5 μL of 1M Tris Base and shaken for 1 min. Glutamate detection reagent was prepared as described in the kit (Promega, Madison, WI, USA, #J7021) with the total volume per sample selected as 12.5 μL. Equal volumes of sample and glutamate detection reagent were added to 384-well white plate (Corning Inc., Corning, NY, USA, #3767). Luminescent signals were detected and quantified using the PerkinElmer Victor^TM^ X4 (PerkinElmer, Waltham, MA, USA).

### 2.8. Intracellular GSH/GSSG Assay

Two 96-well plates were seeded for each cell line at 1 × 10^4^ cells/well density, one for the assay and the other for cell number normalization. After 24 h from seeding, DMSO (vehicle) and 1E5 treatments were added, and the plates were incubated for another 48 h. The total glutathione lysis reagent and oxidized glutathione lysis reagent were prepared as described in GSH/GSSG-Glo^TM^ kit (Promega, Madison, WI, USA, #V6611). After the incubation, plates were washed with PBS twice, and 30 μL of the lysis reagents were added (one set for the total glutathione, and another for oxidized) and shaken for 5 min. A total of 25 μL of lysate was collected on to a 384-well white plate (Corning Inc., Corning, NY, USA, #3767). Freshly prepared 25 μL of luciferin-generating reagent was added and incubated for 30 min. After generating luciferin, 50 μL of luciferin detection reagent was added to each well and incubated for another 15 min. Signals were detected using the PerkinElmer Victor^TM^ X4 plate reader.

### 2.9. ROS Levels

Similar to GSH measurements, two 96-well plates were seeded at 1 × 10^4^ cells/well density for each cell line. DMSO (vehicle) and 1E5 were used as treatments. Incubation times were same as in Section 2.9. At 24 h of incubation, the media was replenished and supplemented with 20 μL of H_2_O_2_ substrate. The plates were then incubated for 6 h at 37 °C. After completion, 50 μL of (media + H_2_O_2_) substrate was pipetted to a 96-well white plate (Corning Inc., Corning, NY, USA, #4517). Freshly prepared 50 μL of ROS-Glo^TM^ (Promega, Madison, WI, USA, #G8820) solution was added and incubated for 20 min. After 20 min, luminescence was read using the PerkinElmer Victor^TM^ X4.

## 3. Results

### 3.1. Expression of LXR and LXR Target Genes in Clinical Samples

Prior to determining the potential role of LXRs and their ligands in breast cancer metabolism, we first examined LXR expression in patient samples. Analysis of RNA-seq data from the TCGA database showed that breast tumors in general showed significantly higher LXRβ expression as compared to normal breast tissues (see Figure 1A). Comparative analysis across different breast cancer subtypes (normal-like, luminal A, luminal B, HER2-plus, and basal-like) also indicated higher LXRβ expression in breast tumors. The opposite was observed for LXRα, with higher LXRα expression in the normal tissue as compared to tumors of all subtypes. Relatedly, we also examined expression levels of known LXR target genes and found that SREBP1c, ACC, and SCD transcript levels were elevated in the tumor tissues, while ABCA1 levels were decreased (see Figure 1B). FASN transcript levels showed no statistically significant differences. Taken together, these results indicate that LXR expression is elevated in breast cancers, and LXRβ appears to be the LXR subtype involved in potential metabolic functions in cancer cells.

### 3.2. Treatments with 1E5 Reduced Breast Cancer Cell Proliferation/Survival

Three breast cancer cell lines (MCF-7, MCF-7-TamR, and MDA-MB-231) were selected to determine the effects of modulating LXR activity with 1E5 in different types of breast cancers. MCF-7 and MCF-7-TamR are both hormone receptor-positive luminal A breast cancer cells. MCF-7-TamR (generated by Bawa-Khalfe and Khan) was derived from MCF-7 as a model for endocrine therapy-resistant breast cancers [18]. MDA-MB-231 cells is a commonly used cell model for triple-negative breast cancers. Treatments with synthetic LXR agonist GW3965, previously shown to have inhibitory activity in breast cancer cells, moderately reduced cell viability as compared to vehicle (DMSO)-treated controls in a concentration-dependent manner in both MCF-7 and MDA-MB-231 cells in tetrazolium salt-reduction MTT assays (see Figure 2A). MCF-7-TamR cells were not affected by GW3965 treatment. Notably, 1E5 treatment significantly reduced cell viability in all three cell lines in a concentration-dependent manner as shown in Figure 2A. To further characterize the inhibitory effects of 1E5, data across different treatment concentrations were plotted and IC50 calculated for the three cell lines used in these studies (see Figure 2B). MCF-7-TamR cells were the most sensitive to 1E5 treatment (IC50 = 7.38 μM), followed by MDA-MB-231 cells (IC50 = 7.74 μM) and MCF-7 cells (IC50 = 8.43 μM). The inhibitory effects of 1E5 were additionally validated and characterized in clonogenic assays (see Figure 2B). Treatments with GW3965 reduced the number of colonies as compared to the DMSO controls, and treatments with 1E5 essentially blocked colony formation in all three cell lines tested. These findings suggest that LXRs play critical roles in breast cancer cell proliferation and survival, and modulating their activity with small molecule ligands may provide a targeted approach for inhibiting tumor growth.

### 3.3. Novel LXR Ligand 1E5 Functions as an Inverse Agonist and Disrupts LXRβ Protein Expression

Initial characterization of 1E5 in pancreatic cancer cells indicated that it functions as a LXR inverse agonist and “degrader” which significantly reduced LXRβ protein levels. To determine if 1E5 has the same mechanisms of action in breast cancer cells, we measured mRNA and protein levels following ligand treatments. In all cell lines examined, LXRβ transcript levels, the abundantly expressed isotype in breast tissue, were shown to be downregulated upon 1E5 treatments (see Figure 3A). In contrast, there was a slight increase (MCF-7 cells) or no significant change in LXRβ transcript levels upon synthetic agonist GW3965 treatment. The effects of novel ligand 1E5 on LXRβ expression were further characterized by Western analysis following treatments (Figure 3B). LXRβ protein levels were significantly decreased in MCF7-TamR and MDA-MB-231 following 1E5 treatment. Interestingly, endocrine-therapy resistant MCF-7-TamR cells showed greater reduction in LXRβ protein level as compared to the parental MCF-7 cells.

A number of LXR target genes have been identified, and they function as key regulators of lipid and cholesterol metabolism (SREBP1c, ABCA1, ACC, FASN, and SCD1). To determine the effects of ligand treatment on LXR activity, mRNA levels of these genes were analyzed using qPCR (Figure 3C). As expected, treatments with LXR agonist GW3965 increased target gene expression. On the other hand, cells treated with 1E5 exhibited downregulation of transcript levels. These results indicate that, similar to their activity in pancreatic cancer cells, 1E5 functions as an LXR inverse agonist in breast cancer, likely through both inhibitory effects on LXR activity and by downregulating LXR protein levels.

### 3.4. Glutamate Metabolism Is Disrupted by LXR Ligand 1E5

During periods of high cellular activity, glycolysis is inadequate for cell survival. Therefore, to cope with the high metabolic stress, tumor cells utilize glutaminolysis to meet their metabolic demands [19]. Glutaminase (GLS1) facilitates the conversion of glutamine to glutamate, a crucial metabolite which contributes to several metabolic mechanisms including the TCA cycle, nucleic acid production, amino acid synthesis, and mTOR signaling activation [20]. To characterize the impact of 1E5 on glutaminolysis, we measured intracellular glutamate levels following ligand treatment (see Figure 4A). Cells were also treated with bis-2-(5-phenylacetamido-1,3,4-thiadiazol-2-yl)ethyl sulfide (BPTES), a small molecule GLS1 inhibitor or BPTES in combination with 1E5. BPTES treatments at 10 μM lowered glutamate levels in all cell lines. Treatments with 1E5 downregulated glutamate levels in a concentration-dependent manner in MCF-7 and MDA-MB-231 cells, and 1E5 showed activity comparable to or greater than BPTES. MCF7-TamR cells exhibited higher base levels of intracellular glutamate as compared to the other cell lines utilized in the study, and 1E5 treatment at 5 μM reduced glutamate levels but had no effect at the higher concentration. The combination of 5 μM BPTES and 5 μM 1E5 showed synergistic effects (greater than expected activity of adding the effects of each single treatment). In addition to measuring their effects on intracellular glutamate levels, we also determined the impact of inhibiting glutaminolysis with 1E5 and BPTES on cell viability using MTT assays (see Figure 4B). As been shown previously (see Figure 2A), 1E5 disrupts cell viability in a concentration-dependent manner. BPTES treatment had no effects on MCF-7 and MCF-7-TamR viability, and only modest effects in MDA-MB-231 cells. Combination treatments only showed slight increases in inhibitory effects in MCF-7 cells, and no additional effects than what was observed in 1E5-only treatments in MCF-7-TamR cells. In MDA-MB-231 cells, there is an additive effect on cell viability when treated with combination of BPTES and 1E5. These results provide evidence that 1E5 is a glutaminolysis inhibitor, although its ability to inhibit glutaminolysis alone is not sufficient to disrupt breast cancer cell proliferation and survival given the modest effects of BPTES.

### 3.5. Expression of Glutaminolysis Genes and Redox Homeostasis Are Disrupted by 1E5

Multiple genes are involved in glutaminolysis, including genes which encode GLS, glutamic-oxaloacetic transaminases 1 (GOT1) and 2 (GOT2), glutamate dehydrogenase 1 (GLUD1), and solute carrier family 7 member 11 (SLC7A11). GOT1 and GOT2 are responsible for the maintenance of cytosolic and mitochondrial oxaloacetate and aspartate equilibrium. GLUD1 converts glutamate to alpha-ketoglutarate (α-KG) and contributes to the TCA cycle [21]. SLC7A11 is an antiporter that exports glutamate while importing cysteine [22]. We carried out qPCR analysis of the transcript levels of these glutaminolysis genes following treatments with the vehicle (DMSO), synthetic agonist GW3965, and 1E5. GLS1 expression was downregulated following 1E5 treatments in all three cell lines (see Figure 5A). GOT1, GOT2, and GLUD1 transcript levels were also downregulated by 1E5, suggestive of possible disruption of TCA cycle anaplerosis. SLC7A11 transcript levels are significantly upregulated in MCF-7 and MCF7-TamR cell lines, providing an additional mechanism for the observed decrease in intracellular glutamate levels.

To further determine the importance of glutamine metabolism in breast cancer cells, we tested their glutamine dependence by culturing them in culture media with and without glutamine supplementation and in the presence and absence of 1E5. Cells deprived of exogenous glutamine showed significantly lower cell numbers as compared to their glutamine-supplemented counterparts (see Figure 5B). Treatments of breast cancer cells with 1E5 in culture media containing glutamine decreased cell viability as compared to vehicle treated controls. The inhibitory effects of 1E5 on cell viability were largely abrogated in the absence of glutamine supplementation, and these observations point to the key role of glutamine metabolism in breast cancers and potentially in mediating the effects of 1E5.

Intracellular glutamate is a precursor for glutathione, a major cellular antioxidant which neutralizes reactive oxygen species (ROS) and thus helping to maintain cellular redox equilibrium [23,24]. To follow the consequence of glutaminolysis inhibition by 1E5, breast cancer cells were assayed for reduced and oxidized glutathione ratios and ROS levels following ligand treatment. Treatments with 1E5 modestly decreased the pool of reduced glutathione and increased the amount of oxidized glutathione, indicative of oxidative stress in MCF-7 cells, and ligand treatment greatly increased oxidative stress (decreased GSH/GSSG ratio) in MCF-7-TamR and MDA-MB231 cells (see Figure 5C). These effects on glutathione ratios are reflected in the increases in ROS levels following treatments with 1E5 (see Figure 5D).

### 3.6. Expression Levels of Glutaminolysis Genes Are Elevated in Breast Tumors and Are Associated with Breast Cancer Patient Survival

To determine the potential clinical relevance of the glutamine metabolism genes in breast cancers, we examined their expression in the patient samples from the TCGA database. GLS1, GOT1, GOT2, GLUD1, and SLC7A11 are expressed at higher levels in tumors as compared to normal breast tissues (see Figure 6A). We then divided patients into high and low-expressing groups based on transcript levels of each glutamine metabolism gene and performed Kaplan–Meier survival analysis. Patients with higher expression levels of these genes showed lower overall survival probability than those with lower expression levels (see Figure 6B). These results strongly suggest that the biochemical pathways and gene networks involved in glutamine metabolism and targeted by the novel LXR ligand 1E5 play key roles in disease progression and response to therapies in breast cancers.

## 4. Discussion

In this study, we have shown that *LXRβ* transcripts are overexpressed in breast tumors across all subtypes, and inhibition of LXR expression and activity by the novel inverse agonist 1E5 also potently inhibited the viability of tumor cells from luminal A and triple-negative breast cancers. Breast cancers are characterized by variable metabolic reprogramming among disease subtypes [15]. Changes in glutamine metabolism have been demonstrated to play key roles in tumorigenesis and drug resistance [25,26,27]. Specifically, it has been reported that triple-negative breast cancers have greater glutamine dependence and tend to overexpress GLS1 [17,28,29]. Therefore, targeting glutamine metabolism and gluaminolysis, perhaps through the novel approach of modulating LXRs by ligands such as 1E5, is a promising targeted therapeutic strategy for these breast cancers which currently lack targeted therapeutic options [30]. While luminal A breast cancers can largely be treated with endocrine therapies using selective estrogen receptor modulators (SERMs) or aromatase inhibitors which block local estrogen production, approximately 40% of patients either do not respond to these therapies or develop resistance over time [31,32]. Intriguingly, 1E5 treatments were effective against MCF-7-TamR cells which were derived from luminal A MCF-7 cells that have developed resistance to the commonly used SERM tamoxifen after prolonged exposure in culture [18]. In fact, these resistant cells had the lowest 1E5 IC50 among the three cell lines tested, and this suggests their greater sensitivity to metabolic disruption by 1E5. These findings await further validation and characterization in vivo in relevant animal models. Mechanistically, glutamine metabolism, particularly dependency on exogenous glutamine, appears to play critical roles in breast cancer cell survival and proliferation, and in mediating the inhibitory effects of the novel LXR ligand 1E5 [15]. Although glutaminolysis is disrupted by 1E5, targeting this mechanism was not sufficient to inhibit proliferation and survival, as evidenced by no or very little effect of the GLS1 inhibitor BPTES. It is possible that other pathways related to glutamine metabolism, such as those involved in nucleotide or fatty acid biosynthesis, and required for tumor cell proliferation and viability are also targeted by 1E5. Another possibility is that inhibition of glutaminolysis synergizes with the non-metabolic mechanism targeted by LXRs or 1E5 specifically. Transcriptomic and metabolomic studies may shed light on these and other mechanisms of action, and provide further insight into the roles of LXRs and their ligands in normal and cancerous tissues.

## Figures and Tables

**Figure 1 biomolecules-13-00345-f001:**
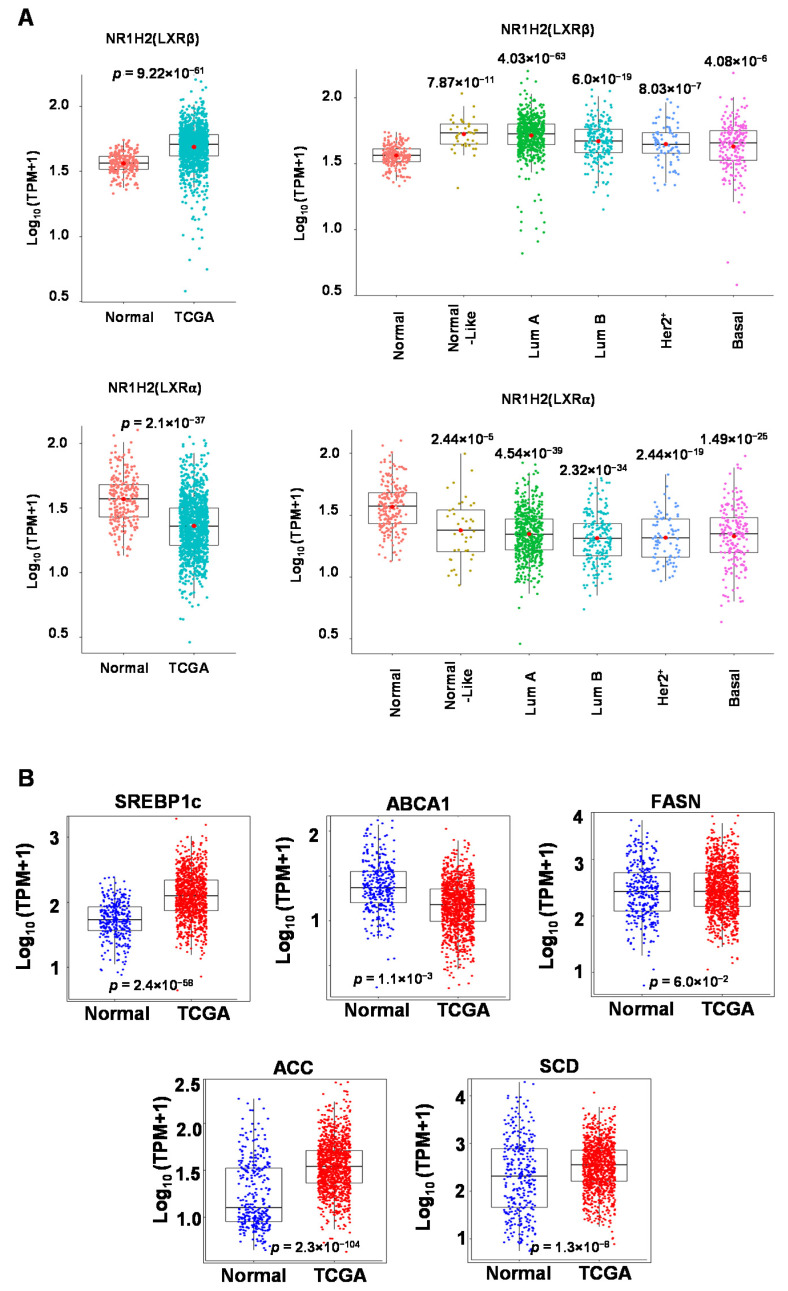
LXRβ and LXR target genes are overexpressed in breast tumors. (**A**) LXRα and LXRβ transcript levels were examined in the TCGA dataset and compared to their expression in GTEx normal breast tissues. Gene expression levels were also compared between breast cancer subtypes (normal-like, luminal A, luminal B, HER2+, and basal-like) and normal breast tissues. (**B**) Known LXR target genes SREBP1c, ABCA1, ACC, SCD, and FASN expression levels were similarly analyzed. Welch’s two-sample T-test was used for all statistical analyses.

**Figure 2 biomolecules-13-00345-f002:**
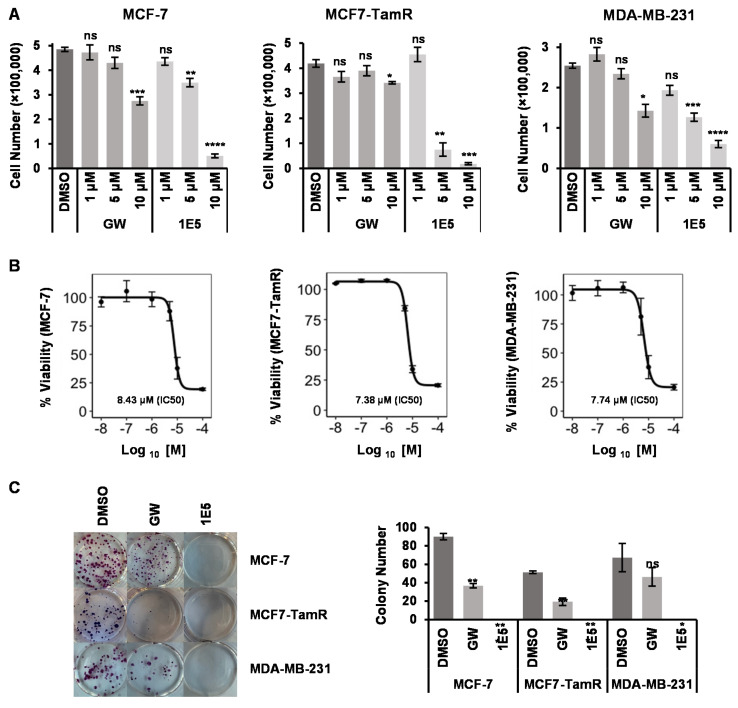
Novel LXR inverse agonist 1E5 disrupts cell proliferation in breast cancer cells in a concentration-dependent manner. (**A**) Trypan blue exclusion assay of breast cancer cell lines MCF-7, MCF-7-TamR, and MDA-MB-231upon incubation for 72 h with three different ligand concentrations (1 μM, 5 μM and 10 μM). (**B**) MTT assays of breast cancer cells after treatment with different 1E5 concentrations for 72 h (0.01 μM, 0.1 μM, 1 μM, 5 μM, 10 μM, 100 μM). These data were used to calculate the IC50 value of 1E5 in all three cell lines. (**C**) Left: colony formation pictures for MCF-7, MCF-7-TamR, and MDA-MB-231 with DMSO (vehicle), GW, and 1E5. Right: number of colonies observed in each condition. All the data represented here consist of biological triplicates (*n* = 3). Standard error was plotted in sets of data as necessary. Statistical significance was determined by Student’s *t*-test (two-tail, two-sample equal variance), where * *p* < 0.05, ** *p* < 0.01, *** *p* < 0.001, **** *p* < 0.0001 and not significant (ns) *p* > 0.05.

**Figure 3 biomolecules-13-00345-f003:**
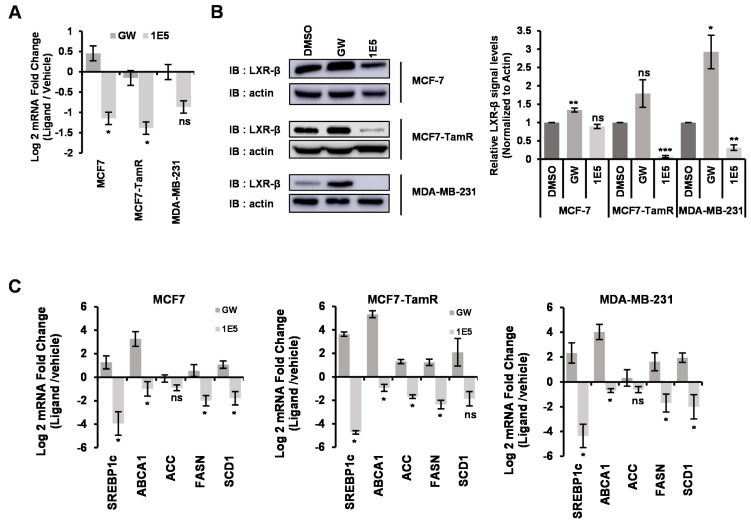
1E5 disrupts LXRβ transcription and translation and acts as an inverse agonist. (**A**) LXRβ transcript levels were reduced in breast cancer cell lines upon 48 h of 1E5 (10 μM) treatments. (**B**) Western analysis of LXRβ proteins following 1E5 treatments showed decreased receptor levels. Each well was loaded with 25 μg of total protein after 48 h of treatments with DMSO, GW, and 1E5. Blots have been tagged for LXRβ and β-actin. Densitometry data that normalized to β-actin were used to calculate the statistical significance among treatments (**C**) 1E5 treatments decreased expression of known LXR target genes SREBP1c, ABCA1, ACC, FASN, and SCD1. RNA was extracted after treatments of DMSO (10 μM), GW (10 μM), and 1E5 (10 μM) for 48 h. All the data represented here consist of biological triplicates (*n* = 3). Standard errors are plotted in the bar graphs. Statistical significance was determined by Student’s *t*-test (two-tail, two-sample equal variance), where * *p* < 0.05, ** *p* < 0.01, *** *p* < 0.001 and not significant (ns) *p* > 0.05.

**Figure 4 biomolecules-13-00345-f004:**
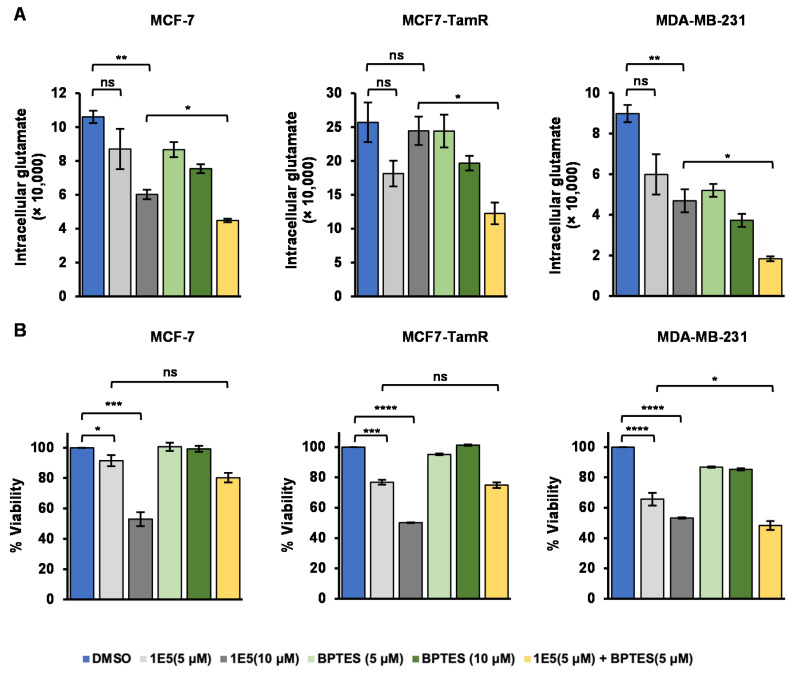
Treatments with 1E5 downregulate the intracellular glutamate levels. (**A**) intracellular glutamate levels were determined by detecting levels of luminescence produced upon the treatments with DMSO, 1E5 (5 μM), 1E5 (10 μM), BPTES (5 μM), BPTES (10 μM) and 1E5 (5 μM) + BPTES (5 μM for 48 h. Luminescence signals were normalized to cell numbers from MTT data. (**B**) MTT assay data used here to detect different levels of cell proliferation with the same set of treatments. All the data represented here consist of biological triplicates (*n* = 3). Standard errors are shown on the bar graphs. Statistical significance was determined by Student’s *t*-test (two-tail, two-sample equal variance), where * *p* < 0.05, ** *p* < 0.01, *** *p* < 0.001, **** *p* < 0.0001 and not significant (ns) *p* > 0.05.

**Figure 5 biomolecules-13-00345-f005:**
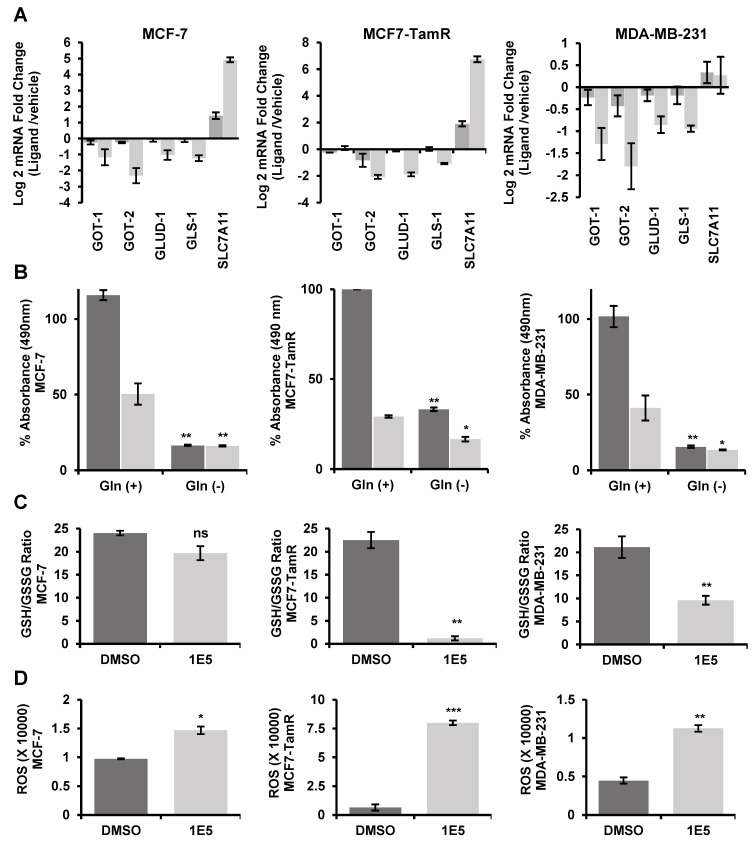
Novel LXR ligand 1E5 induces oxidative stress in breast cancer cells, disrupting glutaminolysis. (**A**) Log2 mRNA expression of known glutaminolysis-related genes GOT1, GOT2, GLUD1, GLS1, and SLC7A11. RNA was extracted after 48 h of ligand treatments. (**B**) MTT analysis of MCF-7, MCF7-TamR, and MDA-MB-231 cells cultured in glutamine-supplemented and stripped media. Cells were treated for 72 h before the MTT assay. (**C**) The ratio of reduced to oxidized (GSH/GSSG) glutathione levels upon 48 h of treatments with DMSO and 1E5. (**D**) Levels of ROS determined as the H_2_O_2_ luminescence upon treatments with DMSO and 1E5 in MCF-7, MCF7-TamR, and MDA-MB-231 cell lines for 48 h. The data presented consist of biological triplicates (*n* = 3). Standard error has been plotted in necessary sets of data. Statistical significance was determined by Student’s *t*-test (two-tail, two-sample equal variance), where * *p* < 0.05, ** *p* < 0.01, *** *p* < 0.001, and not significant (ns) *p* > 0.05.

**Figure 6 biomolecules-13-00345-f006:**
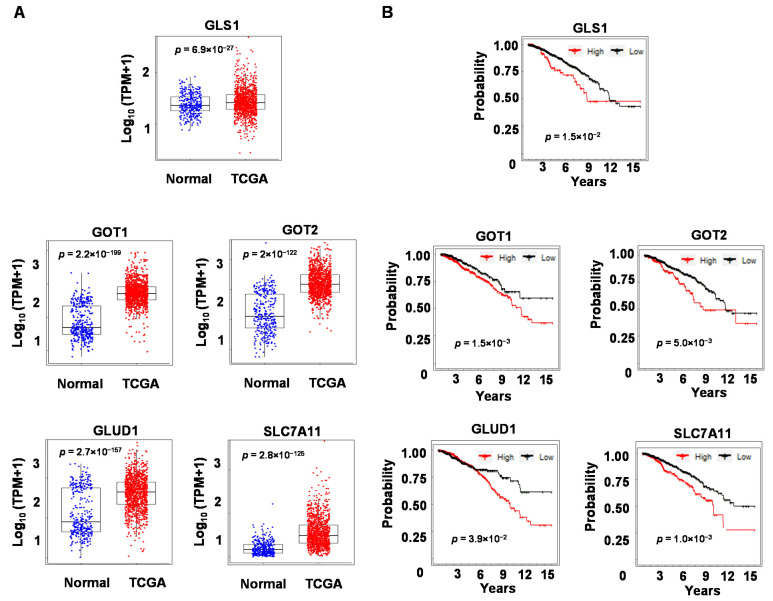
Glutaminolysis genes are upregulated in breast cancer patient samples and associated with lower survival. (**A**) Expression of genes in the glutaminolysis pathway was compared between GTEx normal breast tissue and patients in the TCGA-BRCA cohort using Welch’s two-sample *t*-test. (**B**) Survival analysis of patients with high vs. low expression of these genes was conducted, and the log-rank *p*-values are reported.

## Data Availability

TCGA data presented in these studies are available on publicly available databases.

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
