# Peer review of "Liver X Receptor Inverse Agonist GAC0001E5 Impedes Glutaminolysis and Disrupts Redox Homeostasis in Breast Cancer Cells"

_biomolecules, 2023, doi:10.3390/biom13020345_

Round 1

Reviewer 1 Report

Premaratne et al describe the effects of a novel LXR inverse agonist on glutamine and redox homeostasis in differents breast cancer cell lines. Generally, the report is quite interesting and assess pertinent questions. 

Majors issues for consideration

Statistiques are missing in some figures (figures 2, 3, 4, 5A). Without proper statistics it is difficult to assess the validity of the conclusions.

The value of the western blot data is questionable given that they represent a single run. Regardless of whether apparent density differences are substantial, it is not possible to draw any conclusion from a single blot and without proper statistics.

Minor issues

Line 153: "RIPA suplimented" should be replace by "RIPA supplemented".

Line 286: "inhibitory effect on LXR" should be completed as "inhibitory effect on LXR activity"

In figure 2, it is very surprising that the GW compound has no effect on TamR cell viability but a strong effect on colony formation. The inverse is true as well for the two other cell lines where GW has effect on cell viability but not so much on colony formation. Does the author as any explanation for this observation ?

Reviewer 2 Report

  1. Novelty of the study:  the use of IE5 in breast cancer is novel.  Authors should refer to the seminal work by Falveny et al. (Cancer Cell 2015, Pages 42-56) on LXR inverse agonists and Warburg effect in cancer, which is very relevant to this work.
  1. The inhibition of cell proliferation/survival by IE5 is impressive, which I think has to be validated in vivo. 
  2. The finding that MCF7_TamR showed best results is very interesting and needs further examination and discussion as understanding and tackling endocrine resistance is a very hot topic in breast cancer. The easiest way to do that is to provide RNA seq data for these cells compared to their parent MCF7 and their IE5 treated counterparts.
  1. The presented data is clean and support the conclusion, however, I am not sure if the amount of data is sufficient to meet journal standards. One third of data (3.1 and 3.6) are analyses of publicly available databases which takes a few days to generate. The rest are proliferation/survival, qPCR and western which all are kind of simple experiment. I believe if the authors provide data as suggested in previous comments, manuscript will be much stronger and informative.

Round 2

Reviewer 1 Report

The authors have answered most of the comments from the previous review. There is still 2 points that should be corrected.

Line 276-278: The authors claimed that "LXRβ protein levels were decreased in each cell line but most profoundly in MCF7-TamR and MDA- MB-231 following 1E5 treatment." I don't agree with this assertion which is not supported by the data. In figure 3B, E15 treatment has no significant effect in MCF7 cell line.

Figure 4: the authors have modified the legend of the figure to include the statisticals analysis that has been asked but they do not appear on the figure itself. Nevertheless, the text still described increase/decrease of glutamate levels. Figure should be modified to include the results obtained with the statistical tests and the conclusions as well if there is no statistical effect.

Author Response

Line 276-278: The authors claimed that "LXRβ protein levels were decreased in each cell line but most profoundly in MCF7-TamR and MDA- MB-231 following 1E5 treatment." I don't agree with this assertion which is not supported by the data. In figure 3B, E15 treatment has no significant effect in MCF7 cell line.

Response: Thank you for pointing out this discrepancy between the results of the statistical analysis and our original descriptions. We have revised the results section to state that "LXRβ protein levels were significantly decreased in MCF7-TamR and MDA-MB-231 following 1E5 treatment." MCF-7 has been omitted.

Figure 4: the authors have modified the legend of the figure to include the statisticals analysis that has been asked but they do not appear on the figure itself. Nevertheless, the text still described increase/decrease of glutamate levels. Figure should be modified to include the results obtained with the statistical tests and the conclusions as well if there is no statistical effect.

Response: We apologize for not including the revised Figure 4 with the results of the statistical test added to the figure in our previous resubmission. The revised figure has been added to the manuscript, and the results described in the text are supported by the figure.